# Pandemic Clones of CTX-M-15 Producing *Klebsiella pneumoniae* ST15, ST147, and ST307 in Companion Parrots

**DOI:** 10.3390/microorganisms10071412

**Published:** 2022-07-13

**Authors:** Yamê Miniero Davies, Marcos Paulo Vieira Cunha, Milena Dropa, Nilton Lincopan, Vasco Túlio Moura Gomes, Luisa Zanoli Moreno, Maria Inês Zanoli Sato, Andrea Micke Moreno, Terezinha Knöbl

**Affiliations:** 1School of Veterinary Medicine and Animal Science, University of São Paulo, Av. Prof. Dr. Orlando Marques de Paiva, 87, Cidade Universitária, São Paulo 05508-270, SP, Brazil; yamedavies@gmail.com (Y.M.D.); cunha.mpv@gmail.com (M.P.V.C.); vaskotulio@yahoo.com.br (V.T.M.G.); luisa.moreno@usp.br (L.Z.M.); morenoam@usp.br (A.M.M.); 2Environmental Health Department, School of Public Health, University of São Paulo, Avenida Dr. Arnaldo, 715, Cerqueira César, São Paulo 05508-270, SP, Brazil; milenadropa@gmail.com; 3Department of Microbiology, Institute of Biomedical Sciences, Universidade de São Paulo, Av. Prof. Lineu Prestes, 2415-Butantã, São Paulo 05508-000, SP, Brazil; lincopan@usp.br; 4São Paulo State Environment Agency (CETESB), Av. Prof. Frederico Hermann Jr. 345. Pinheiros, São Paulo 05459-010, SP, Brazil; misato@sp.gov.br

**Keywords:** psittacine birds, virulence, ESBLS, antimicrobial resistance, zoonotic pathogens

## Abstract

Psittacine birds are commonly kept as companion birds and the maintenance of these birds in captivity may represent a zoonotic risk and contribute to the propagation of multidrug-resistant and β-lactamase extended-spectrum (ESBLs)-producing pathogens. This study aimed to identify and characterize strains of the *Klebsiella* *pneumoniae* complex isolated from diseased psittacine birds, determining virulence and resistance profiles. *K. pneumoniae* strains were isolated from 16 birds (16/46). All strains carried more than three virulence genes, with a high frequency of *fimH* and *kpn* (93.75%), *uge* (87.52%), and *irp*-2 (81.25%) genes. The antimicrobial susceptibility revealed that 3/16 strains were ESBL producers. Genomic analysis revealed that CTX-M-15-positive strains belonged to sequence types (STs) ST15, ST147, and ST307, characterized as international clones associated with outbreaks of healthcare-associated infections (HAIs) worldwide.

## 1. Introduction

Members of the *Klebsiella pneumoniae complex* are adapted to survive in environments and have a broad ecological range, including wastewater, human, and domestic animals. The greatest concern about *K. pneumoniae* is associated with the potential to amplify and spread clinically important antimicrobial resistance genes (AMR) [1]. *K. pneumoniae* is highlighted as an opportunistic pathogen that causes urinary tract infection, pneumonia, wound infections, hemorrhagic colitis, and sepsis [2]. Symptomatic and asymptomatic individuals can function as a reservoir of an agent, after colonization of the nasopharyngeal and gastrointestinal tract, and represent a risk of transmission of multidrug-resistant strains [1]. Specifically, some international clones of *Klebsiella pneumoniae* have become a global public health issue due to virulence and/or multidrug-resistant profiles [1]. In this regard, the most common resistant lineages of *K. pneumoniae* have been associated with the production of extended-spectrum beta-lactamase (most CTX-M-15 variants) and/or KPC-type carbapenemases, being categorized as critical priority pathogens by WHO (2020) [3]. On the other hand, although clones belonging to clonal group CC58 have been predominant in healthcare-associated infections (HAIs) worldwide, novel STs have emerged as high-risk clones, such as ST15, ST147, and ST307, which are associated with high human mortality [4,5,6].

Since most surveillance studies focus on the impact of human nosocomial infections, the pandemic clones of *K. pneumoniae* have been reported in companion animals, such as dogs, and cats [1,7,8,9]. Chen et al. (2021) evaluated the dynamics of transmission of *K. pneumoniae* between dogs and their owners. The authors showed a clonal diverse population of *K. pneumoniae* in the gastrointestinal tracts of healthy humans, but some dominant clones were present in pets that shared the same house, including mammals and reptile pets [10].

The role of the microbial community of wild birds on the trafficking of AMR genes in one health approach remains unknown [7,8,9]. Free-living psittacine birds are colonized by Gram-positive bacteria, but they acquire the Gram-negative microbiota during captivity in conservation and rescue centers. There they can spread resistant pathogens from stool. Silva et al. (2021) reported the presence of KPC-producing *E. coli* in Psittaciformes rescued from trafficking in Brazil. The authors highlighted the risks associated with reintroduction or domestic breeding of these birds [11]. In a previous study, we reported the importance of virulent strains of *K. pneumoniae* isolated from parrots and passerines seized from illegal wildlife trade, which carried plasmid-mediated quinolone resistance, SHV-1, SHV-11, and *bla*TEM-1 genes [9].

In this study, we report the occurrence and characterization of international clones of CTX-M-15-positive *K. pneumoniae* causing respiratory diseases in companion parrots, highlighting the zoonotic and anthropozoonotic potential of this critical priority pathogen, within a One Health perspective.

## 2. Materials and Methods

### 2.1. Animals and Bacterial Strains

This study investigated 16 strains of *K. pneumoniae* isolated from psittacine birds in São Paulo, Brazil (Ethics Committee of São Paulo University, CEUA 5174111215 and SISBIO: 46561-2). *K. pneumoniae* strains were isolated from clinical samples from 46 companion parrots presenting respiratory symptoms (sinusitis and pneumonia). Clinical signs were nasal discharge, edema periocular, caseous in the nostril, and dyspnea. Samples of organs, secretion, and excretions were collected with sterile swabs and forwarded to the Avian Medicine Laboratory, School of Veterinary Medicine and Animal Science, University of São Paulo, Brazil. The species of psittacine assessed included *Amazona aestiva* (*n* = 30), *Amazona amazonica* (*n* = 3), *Amazona xanthops* (*n* = 1), *Anodorhynchus hyacinthinus* (*n* = 3), *Psephotus haematonotus* (*n* = 1), *Agapornis* spp. (*n* = 1), *Psittacara leucophthalmus* (*n* = 2), and *Nymphicus hollandicus* (*n* = 5).

The swabs were incubated in brain heart infusion broth (Difco™) at 37 °C for 18 h and cultured on MacConkey agar (Difco™) at 37 °C for 24 h. Selected colonies were subjected to species identification by MALDI-TOF MS (Matrix-Assisted Laser Desorption/Ionization Time-of-Flight Mass Spectrometry) [12].

### 2.2. Antimicrobial Susceptibility Tests

The antimicrobial resistance was determined by disk diffusion [13] for nalidixic acid, enrofloxacin, levofloxacin, amikacin, gentamicin, tobramycin, amoxicillin-clavulanic acid, cefotaxime, cefoxitin, chloramphenicol, sulphonamides, sulfamethoxazole-trimethoprim, tetracycline, and imipenem. *Escherichia coli* ATCC25922 was used as a control. The ESBL production was screened by the double-disc synergy test (DDST) [13]. Minimal inhibitory concentrations (MICs) to cefotaxime, cefepime, ceftazidime, and ceftiofur were assessed by agar dilution method on Mueller Hinton agar plates, as recommended by the Clinical and Laboratory Standards Institute [13].

### 2.3. Virulence and Genotypic Profile of Klebsiella pneumoniae Strains

The virulence profile was investigated by PCR for screening of *K. pneumoniae* virulence genes *iroN, irp-2, rmpA, magA, kfu, uge, kpn, mrkD, fimH, cc258, allS*, K1 and K2 capsular polysaccharides [9].

The genotypic profile was performed by single-enzyme amplified fragment length polymorphism (AFLP), with HindIII restriction endonuclease (5′ AAGCTT 3′ 3′ TTCGAA 5′) (Invitrogen, Inc., Waltham, MA, USA) [14]. BioNumerics (version 7.6 AppliedMaths, Sint-Martens-Latem, Belgium) was employed for cluster analysis of Dice similarity using the Unweighted Pair-Group Method Using Arithmetic Average (UPGMA) method. A cutoff point of 90% was employed for the determination of similar genotypic profiles [15].

### 2.4. Plasmid Analysis

ESBL-producing isolates were selected to demonstrate the transferability of the plasmids. Conjugation was conducted using *Escherichia coli* C600 as a receptor strain. Donor and receptor strains (2:1) were inoculated in 5 mL of Luria Bertani broth (Difco™). Transconjugants were selected after 24 h of incubation at 35 °C, 100 µL of the inoculum was plated in MacConkey agar (Difco™) supplemented with 2000 mg/L of streptomycin and 2 mg/L of cefotaxime. For the isolate carrying both *bla*_CTX-M-15_ and *bla*_CTX-M-8_, ceftazidime 4 mg/mL was used instead of cefotaxime, to select colonies with only the plasmid carrying *bla*_CTX-M-15_ [16]. The plasmid sizes were determined by S1-nuclease PFGE and the plasmids were typed by PCR-based replicon typing and replicon sequence typing of IncF plasmids [17].

### 2.5. Whole Genome Sequencing and Analysis of ESBL-Producing K. pneumoniae Strains

All strains exhibiting ESBL phenotype were selected for whole-genome sequencing. Genomic DNA was extracted and purified from overnight pure cultures using the PureLink Genomic DNA purification Kit (Invitrogen) following the manufacturer’s recommendations. Nextera XT DNA Library kit (Illumina^®^) was used to generate paired-end libraries (2 × 150 bp) according to the manufacturer’s instructions, followed by sequencing in the Illumina NextSeq platform (Illumina, San Diego, CA, USA).

Quality of raw reads was assessed by FastQC (v.0.72), and read trimming was conducted using Trimmomatic (v.0.38.0). Genomes were de novo assembled by SPAdes (v.3.9.0) and annotated with Prokka version 1.13 [18] and NCBI Prokaryotic Genome Annotation Pipeline (PGAP) (https://www.ncbi.nlm.nih.gov/genome/annotation_prok/, accessed on 18 November 2020). Resistance genes, multilocus sequence typing, and plasmids were in silico identified using tools of the Center for Genomic Epidemiology (MLST, Resfinder, Plasmidfinder, pMLST) (http://www.genomicepidemiology.org/, accessed on 18 November 2020). Serotyping of K antigen was determined using the online tool Kaptive (v.0.7.2) [19]. The genetic environment of *bla*_CTX-M_ genes was determined in silico using the ISfinder database (https://isfinder.biotoul.fr, accessed on 27 November 2020) and PCR gap closure when necessary. The Whole Genome Sequencing project was deposited at DDBJ/EMBL/GenBank under the accession number PRJNA832859. Three whole genomes were deposited at GenBank under the accession numbers JALYBQ000000000, JALYBR000000000, and JALYBS000000000.

## 3. Results

Sixteen *Klebsiella* spp. strains were isolated from 46 infected birds, and all colonies were identified as *K. pneumoniae* by MALDI-TOF MS. All strains carried three to five virulence genes, being positive for *fimH* (100%), *kpn* (93.75%), *uge* (87.50%), *irp*-2 (81.25%), *mrkD* (68.75%), and *kfu* (37.50%). Results identified five different virulence profiles, with predominance of *irp2+uge+kpn+mrkD+fimH* (8/16) and *irp2+uge+kpn+kfu+fimH* (5/16) genotypic traits. The SE-AFLP classified these strains into 15 distinct patterns, with a discriminatory index of 0.96. Studied strains were from noncontacting birds, located in various places, and were heterogeneous (Figure 1).

Three strains were ESBL-producers and exhibited a multidrug-resistant (MDR) profile, of which two were positive for the *bla*_CTX-M-15_ gene and one was positive for both *bla*_CTX-M-15_ and *bla*_CTX-M-8_ genes. These strains were selected for whole-genome sequencing. The genomes ranged between 5,622,065 and 5,540,827 bp, which is according to the genome size of *K. pneumoniae* (mgh). Genomic features of these strains are shown in Table 1. In these isolates, it was possible to transfer resistance to third-generation cephalosporins to a transconjugant strain through conjugation assays (Table 2).

MDR phenotype in CTX-M-15-producing *K. pneumoniae* strains was related to genes conferring resistance to phenicols, beta-lactams, aminoglycosides, fosfomycin, quinolones, sulphonamides, sulfamethoxazole-trimethoprim, and tetracyclines. Additionally, chromosomal mutations in the *gyrA* and *parC* genes were responsible for elevated levels of resistance to fluoroquinolones (Table 1).

In silico analysis showed that these strains belonged to ST15, ST147, and ST307. ST15 isolate (Kp58.3) was isolated from the oropharynx of a *Psittacara leucophtalmus* specimen. This isolate presents the *bla*_CTX-M-15_ gene in a ~250 Kb plasmid belonging to K5:A10: B-replicon sequence type (RST) (Table 1 and Table 2). Through BLAST alignments, we found this isolates the presence of the *kpi* operon. This chaperone-usher pili system is related to the ability to produce remarkable adherent phenotypes with abundant fimbriae structures. Our ST15 isolate has a 6628 bp DNA fragment in the genome that encodes seven genes (*kpiA, kpiB, kpiC, kpiD, kpiE, kpiF, kpiG*) and showed 100% similarity to the *kpi* operon of the Kp3380 strain, an MDR ST15 strain *K. pneumoniae* previously isolated from a hospital outbreak in Spain [20].

The strain belonging to ST307 (Kp137) was isolated from a respiratory secretion of the *Amazona*
*aestiva* parrot that died of sepsis. This strain harbors three plasmids, with *bla*_CTX-M-15_ located in an IncFIB(K) plasmid sizing ~145 Kb. On the other hand, the isolate Kp41 was recovered from a nasal secretion of an *Amazona aestiva* parrot and belonged to ST147. Kp41 strain possesses an ~220 kb K9:A-:B-plasmid carrying *bla*_CTX-M-15_, and other plasmids belonging to the IncM1 family, harboring *bla*_CTX-M-8_ (Table 1). The IncM1 plasmid was analyzed in a previous study [21]. The *bla*_CTX-M-8_ gene was inserted in an IS*26*-composite transposon, composed of an IS*10* copy and two IS*26* copies present in that IncM1 plasmid.

In all three strains, the *bla*_CTX-M-15_ gene was found upstream IS*Ecp*1 element, with ORF477 downstream. This module presented intact repeat regions (IRs) of IS*Ecp*1, a classic genetic environment of *bla*_CTX-M-15_. In the isolates belonging to ST15 and ST307, this structure was downstream to a resistance region carrying resistance genes to aminoglycosides (*aph(3″)-Ib, aph(6)-Id*), beta-lactams (*bla*_TEM-1_), and sulphonamides (*sul2*) (Figure 2).

These CTX-M-15-producing *K. pneumoniae* strains showed an MDR phenotype. Using in silico analysis it was possible to correlate with many acquired genes conferring resistance to antimicrobial drugs (Table 1). In addition to the acquired resistance genes, it was possible to determine chromosomal mutations in the *gyrA* and *parC* genes, which confer elevated levels of resistance to fluoroquinolones.

## 4. Discussion

Convergence of virulence and multidrug resistance in *Klebsiella pneumoniae* is considered a major public health concern [22,23]. This species is one of the most common MDR bacteria isolated in nosocomial and community-acquired infections in humans [24]. The reports about infection by *K. pneumoniae* in veterinary medicine have been associated with cats and dogs [7,25,26,27,28], and little is known about the impact of this pathogen in companion and wildlife bird species. In a previous study, colonization of passerine and psittacine seized from illegal trade in Brazil, by *K. pneumoniae*, was reported [9]. In this study, the occurrence of *K. pneumoniae* among psittacine birds was 34.78%, and most of these strains presented virulence factors, including genes associated with fimbriae (*fimH*—93.75%, *fim1*-like *kpn*—93.75% and type 3 fimbriae *mrk*D—62.50%) and siderophores (*irp*-2—81.25% and *kfu*—43.75%). According to Lawlor et al. (2007), yersiniabactin is the most important iron acquisition system of *K. pneumoniae* during in vivo pulmonary infection [29]. Holt et al. (2015) suggested that the acquisition of iron-scavenging systems increases the risk of severe invasive disease and highlighted the public health concern of the yersiniabactin in the ESBL clones, including ST15 *K. pneumoniae* [22].

In this study, none of the strains presented genes *mag*A and *rpm*A, associated with hypermucoviscosity. However, 87.20% were positive for the uridine diphosphate galacturonate 4-epimerase gene (*uge*), which is associated with the integrity of smooth LPS [30].

Among the multidrug-resistant lineages, some clonal groups (CGs) such as CG258, CG307, CG101, CG147, and CG15 are globally disseminated and responsible for an extensive range of infections in humans [2,31]. In this study, three of these STs were found to cause respiratory disease in parrots. In this regard, ST307 has been a pandemic lineage related to CTX-M-15-production, emerging in the middle of the 1990 decade [1]. Genomic studies reveal that ST307 presents a conserved genome, differing only in the mobilome (i.e., plasmids, phages, and resistance genes) [32]. CG307 has some characteristics that favor the colonization of humans and the hospital environment, such as virulence factors and plasmids. Recently, CTX-M-15-positive ST307 strains were implicated in nosocomial outbreaks in Germany and The Netherlands [33,34]. In Brazil, ST307 strains were recovered from wastewater and a urinary tract infection in cats and dogs [7,35,36].

In addition, our study also detected the lineages ST147 and ST15 in psittacine isolates. These lineages of *K.*
*pneumoniae* producing CTX-M-15 have been described in companion animals, including dogs with urinary tract infections and horses [37,38]. Our results reinforce the link of transmission between humans and pets, including companion birds.

The three ESBL-producing strains here studied belong to the CTX-M-15 variant. CTX-M-15 is the most common ESBL enzyme in *Klebsiella pneumoniae* isolated from human nosocomial infections worldwide. Through a wide literature review, Calbo and Garau (2015) showed that the epidemiology of ESBL-producing *K. pneumoniae* changed in the 2000s, and CTX-M-15 has completely replaced other ESBL enzymes, including TEM, SHV, and other CTX-M variants worldwide [39].

One of the reasons for the successful dissemination of CTX-M-15 in hospitals may be the fact that it is an enzyme that has an increased catalytic activity against ceftazidime, a third-generation cephalosporin produced for the treatment of *Pseudomonas* spp. Unlike other worldwide disseminated variants of CTX-M, CTX-M-15 and hybrid-CTX-M-15 enzymes producing bacteria are resistant to ceftazidime.

Plasmids have a key role in the dissemination of CTX-M genes enzymes. IncF and IncN plasmids are associated with the worldwide dissemination of *bla*_CTX-M-15_ [40]. In our study, the gene was present in IncF (K) plasmids. The three strains analyzed presented *bla*_CTX-M-15_ gene upstream IS*Ecp*1 element, with ORF477 downstream, which is the worldwide reported structure [41,42]. These sequences show 100% similarity to the IncF (K) plasmids described in *K. pneumoniae* isolates in Colombia, the United States, and Australia (GenBank access CP024566, CP024516, CP016925, and CP10390). Previous studies in Brazil have identified the *bla*_CTX-M-15_ gene in plasmids of the IncR family, in *K. pneumoniae* isolated from food-producing animals [43].

In our work, the CTX-M-8 variant was inserted in an IS*26*-composite transposon, composed of an IS*10* copy and two IS*26* copies. An identical transposon has been identified in plasmids of the Incl1 and IncM1 families in environmental isolates in Brazil [35], and in chicken meat imported from Brazil [44]. In addition to *bla*_CTX-M-8_, pKp41 M plasmid contained a resistance region composed of Tn*1331*, truncated by a 4,108 pb module containing ISE*cp*-*qnrE1*-*araJ*-*∆ahp*. The *qnrE1* gene, which confers resistance to quinolones, was described in a human clinical strain of *K. pneumoniae* in Argentina [45]. However, the genetic environmental described by Albornoz [45] was not associated with Tn*1331*, as in the pKp41 M plasmid.

## 5. Conclusions

In conclusion, our results showed that international clones of CTX-M-15-positive *K. pneumoniae* can cause infectious diseases in companion birds, highlighting the zoonotic and anthropozoonotic potential of this critical priority pathogen, within a One Health perspective.

## Figures and Tables

**Figure 1 microorganisms-10-01412-f001:**
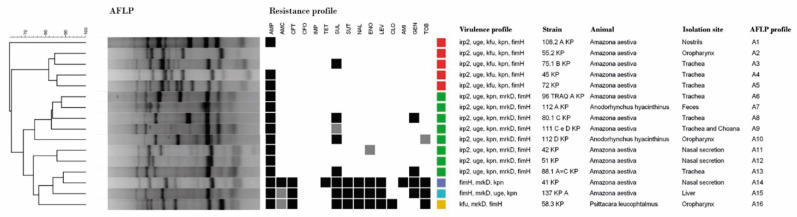
Dendogram showing the relationship among the SE-AFLP patterns of *K. pneumoniae* isolated from companion birds.

**Figure 2 microorganisms-10-01412-f002:**
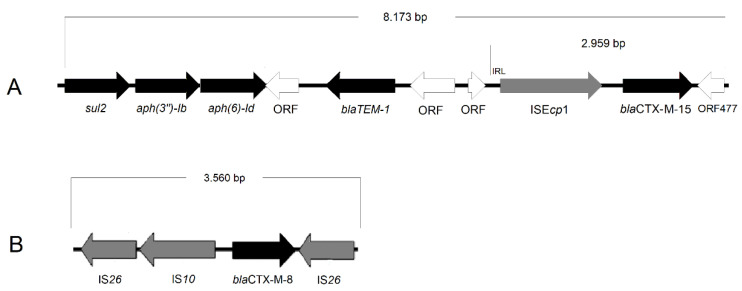
Genetic environment of *bla*_CTX-M-15_ (**A**) and *bla*_CTX-M-8_ (**B**) in ESBL-producing isolates. Black arrows indicate resistance genes, gray arrows indicate genes related to mobile elements, and white arrows indicate open read frames (ORFs) (bp = base pairs).

**Table 1 microorganisms-10-01412-t001:** Characteristics and genomic data of *K*. *pneumoniae* strains infecting companion birds.

Isolates Information	Strains		
	Kp41	Kp58	Kp137
Host	*Amazona aestiva* (nasal secretion)	*Psittacara leucophtalmus* (oropharynx)	*Amazona aestiva* (sepsis)
Codified Regions (CDS)	5548	5351	5436
RNAs	93	96	86
GC Content	57.1%	57.2%	57.1%
*Sequence type*	ST147	ST15	ST307
Plasmids-group (size)	FII(K) (~220 kb) L/M (~70 kb) ColRNAI (3 kb)	FII(K)-FIA-FIB (~250 kb)	FIB (K) (~145 kb) NT (48 kb) ColRNAI (4 kb)
Capsule	K64	K60	K173
	Resistance genes		
Aminoglycosides	*aac(3)-VIa, aph(3″)-Ib, aac(3)-IIa, aadA1, aph(6)-Id*	*aph(3″)-Ib, aac(6′)Ib-cr, aph(6)-Id, aadA2*	*aac(3)-IIa, aph(3″)-Ib, aac(6′)Ib-cr, aph(6)-Id*
Beta-lactams	*blaSHV-11, blaCTX-M-8, blaCTX-M-15, blaTEM-1 A, blaOXA-1, blaOXA-9*	*blaSHV-28, blaCTX-M-15, blaTEM-1 B, blaOXA-1*	*blaSHV-28, blaCTX-M-15, blaTEM-1 B, blaOXA-1*
Quinolones	*qnrE*, *oqxAB, gyrA* (mutation)	*aac(6′)Ib-cr*, *oqxAB, gyrA* (mutation)	*aac(6′)Ib-cr*, *oqxAB*, *qnrB66*, *gyrA* (mutation)
Fosfomycin	*fosA*	*fosA*	*fosA*
Tetracyclines	*tet(A)*	*-*	
Sulfonamides	*sul1, sul2*	*sul1, sul2*	*sul2*
Trimethoprim	*dfrA14*	*dfrA14, dfrA12*	*dfrA14*
Phenicols	*catB4*	*catA1, catB4*	*catB4*

**Table 2 microorganisms-10-01412-t002:** Minimum inhibitory concentration and plasmid sizes of CTX-M-15-producing isolates and respective transconjugants.

Strain	Plasmid Sizes	MIC (mg/L)			
		Cefotaxime	Ceftazidime	Cefepime	Ceftiofur
Kp41	220 kb 70 kb 3 kb	>128	128	>128	>128
Tc-41a	220 kb	>128	32	>128	>128
Kp58	250 kb	>128	128	>128	>128
Tc-58a	250 kb	>128	64	>128	>128
Kp137	145 Kb 48 kb 4 kb	>128	128	>128	>128
Tc-137a	145 Kb	>128	64	>128	>128
EC-C600	-	0.125	0.5	0.125	0.125

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
