# Peer review of "Pandemic Clones of CTX-M-15 Producing Klebsiella pneumoniae ST15, ST147, and ST307 in Companion Parrots"

_microorganisms, 2022, doi:10.3390/microorganisms10071412_

Round 1

Reviewer 1 Report

The paper "Pandemic Clones of CTX-M-15 Producing Klebsiella pneumoniae ST15, ST147 and ST 307 in Companion Parrots" by Davies et al is a descriptive and very straightforward piece. It describes the strains of the K. pneumoniae isolated from diseased companion birds and characterizes their virulence profile. 

The introduction is very short and not a lot is discussed about the relevance of K. pneumoniae in companion birds. 

The representation of the results is very basic and Figure 1 is even difficult to read, I suggest that the authors make other tables or figures to better explain the results. Figure 1 contains many results and should have a better layout.

Author Response

Dear Reviewer

We really appreciate the review of our study. Based on the observation we improved the Introduction (highlighted in yelow). Unfortunately the number of mannuscripts about the relevance of K. pneumoniae in psittacine birds are very restrict and we believe that the publication of this data will serve as a stimulus for new studies in this field. We also included 6 references, one figure and a new table.   

The Figure 1 was generate by Bionumerics software. We believe that the information can be clarified if the size of dendogram is enlarged. We are sending a separate file (high resolution) for the publisher, and the designer can adjust to the page size or create a link to improve the visualization of Fig. 1. Thank you. 

 Suzanne Kane, an english-speaking writer/editor, edited this paper (please see the attachment).   

Best regards

Reviewer 2 Report

About this study:

The study's aim was to identify and characterize strains of the Klebsiella pneumoniae complex isolated from diseased 14 psittacines birds, determining virulence and resistance profiles. K. pneumoniae strains were identified in 16 birds.

Authors ascertained that strains carried more than 3 virulence genes, with a high frequency of four genes: fimH / kpn (93.75%), uge (87.52%), and irp-2 (81.25%). The antimicrobial susceptibility revealed that three of 16 strains were ESBL producers.

Also, the genomic analysis revealed that CTX-M-15-positive strains belonged to sequence types (STs) ST15, ST147, and ST307, considered international clones associated with outbreaks of healthcare-associated infections (HAIs) worldwide.

Hardpoints:

The novelty of the study, with new inputs.

M & M - easy to understand and replicable.

Results and Discussion are well presented.

What to improve:

The introductory part is too short, please develop more introducing additional info, especially about the One Health issue in this case.

Please introduce some statistical elements correlating virulence with the frequency of the main genes ascertained. 

Author Response

Dear Reviewer

We really appreciate the review of our study. Based on the observation we improved the Introduction and included some information about the % of virulence genes (highlighted in yelow). We also included 6 references, one figure and a new table.   

This paper provide a description of virulence genes. Therefore we express the frequency of each gene in % and group the strains according the genotypes. In a previous study (Davies et al., 2016), we had 2 groups of animals (diseased and non-diseased) which allowed for a more detailed statistical comparison. Here we highligted that virulent strains, isolated from sick birds, can carry resistance genes. Some resistant strains belonged to the pandemic lineages (One Health approach, as mentioned). Thanks. 

Suzanne Kane, an english-speaking writer/editor, edited this paper (please see the attachment).   

Best regards

Reviewer 3 Report

Dear Authors!

My comments are in the file

Author Response

Dear Reviewer

We really appreciate the review of our study. Based on the observation we improved the results section (See Lines 145- 147 and L. 174-185) Thanks.

We included  one table and one figure: 

Table 2. Minimum inhibitory concentration and plasmid sizes of CTX-M-15-producing isolates and respective transconjugants.)

Figure 2. Genetic environment of blaCTX-M-15 (A) and blaCTX-M-8 (B) in ESBL-producing isolates...

Also, Suzanne Kane, an english-speaking writer/editor, edited this paper (please see the attachment).   

Best regards

Round 2

Reviewer 2 Report

Dear Authors,

Now I agree with your work to be accepted.

Reviewer 3 Report

Dear Authors!

Thank you for tha corrections. Now, I think, the MS can be accepted.